# Tetraspanin 8 Subfamily Members Regulate Substrate-Specificity of a Disintegrin and Metalloprotease 17

**DOI:** 10.3390/cells11172683

**Published:** 2022-08-29

**Authors:** Miryam Müller, Claire Saunders, Anke Senftleben, Johannes P. W. Heidbuechel, Birgit Halwachs, Julia Bolik, Nina Hedemann, Christian Röder, Dirk Bauerschlag, Stefan Rose-John, Dirk Schmidt-Arras

**Affiliations:** 1Institute of Biochemistry, Medical Faculty, Christian-Albrechts-University Kiel, 24118 Kiel, Germany; 2Tumour Immunology Laboratory, Department of Biosciences and Medical Biology, Paris-Lodron-University Salzburg, 5020 Salzburg, Austria; 3Department of Gynaecology, University Medical Center Schleswig-Holstein, 24105 Kiel, Germany; 4Institute of Experimental Cancer Research, Medical Faculty, University Medical Center Schleswig-Holstein, 24105 Kiel, Germany

**Keywords:** proteases, tetraspanin, TNF, inflammation

## Abstract

Ectodomain shedding is an irreversible process to regulate inter- and intracellular signaling. Members of the a disintegrin and metalloprotease (ADAM) family are major mediators of ectodomain shedding. ADAM17 is involved in the processing of multiple substrates including tumor necrosis factor (TNF) α and EGF receptor ligands. Substrates of ADAM17 are selectively processed depending on stimulus and cellular context. However, it still remains largely elusive how substrate selectivity of ADAM17 is regulated. Tetraspanins (Tspan) are multi-membrane-passing proteins that are involved in the organization of plasma membrane micro-domains and diverse biological processes. Closely related members of the Tspan8 subfamily, including CD9, CD81 and Tspan8, are associated with cancer and metastasis. Here, we show that Tspan8 subfamily members use different strategies to regulate ADAM17 substrate selectivity. We demonstrate that in particular Tspan8 associates with both ADAM17 and TNF α and promotes ADAM17-mediated TNF α release through recruitment of ADAM17 into Tspan-enriched micro-domains. Yet, processing of other ADAM17 substrates is not altered by Tspan8. We, therefore, propose that Tspan8 contributes to tumorigenesis through enhanced ADAM17-mediated TNF α release and a resulting increase in tissue inflammation.

## 1. Introduction

Limited proteolysis is an irreversible post-translational mechanism to control signaling processes. Proteolytic cleavage of ectodomains of transmembrane protein is termed “ectodomain shedding” and is involved in both negative and positive regulation of auto- and paracrine signaling events. Members of the a disintegrin and metalloprotease (ADAM) family are major mediators of ectodomain shedding. They all share a similar domain structure and are characterized by an invariant zinc-binding motif within the catalytic subunit. They are synthesized as an inactive pro-form containing the N-terminal pro-domain that folds back to the catalytic domain. The pro-domain is proteolytically removed in the Golgi, giving raise to the mature form of ADAMs [1,2]. The two homologues ADAM10 and ADAM17 are the best-characterized members of the family. More than 70 substrates have been identified for each protease in vitro [1,2] with a partially overlapping substrate spectrum. While ADAM10 was initially identified as a major sheddase for the Notch receptors, ADAM17 is a major sheddase for tumor necrosis factor (TNF) α, thereby enabling paracrine and systemic TNF α signaling [3,4]. Their disintegrin and membrane-proximal domains were demonstrated to participate in substrate recognition [5] and activation of ADAM17 catalytic activity by membrane outer leaflet phosphatidyl serine [6]. However, the detailed molecular mechanism underlying the regulation of ADAM protease activity and substrate selectivity still remains elusive. Recently, inactive rhomboid protease family members (iRhoms) and tetraspanins have emerged as regulators of ADAM proteases [7,8].

Tetraspanins (Tspans) are a large family of small transmembrane proteins that are evolutionary conserved. To date, 33 members of the Tspan family have been identified in humans and can be grouped into subfamilies based on sequence clustering. Tspans can be found in almost all cell and tissue types. As the name indicates, Tspans span the plasma membrane four times with both N- and C-terminus at the cytoplasmic side [9]. As a consequence, one small extracellular loop (SEL) between transmembrane region 1 and 2 and one large extracellular loop (LEL) between transmembrane region 3 and 4, respectively, is formed. The LEL consists of a constant region that is thought to confer dimerization with tetraspanins (homo- and hetero-dimerization) and a variable region that is supposedly required for interactions with non-tetraspanin partner proteins [9]. Through their ability to interact with one another and with partner proteins, Tspans have the ability to create large dynamic interacting networks within the membrane, also often referred to as the tetraspanin web. These interactions are modulated by membrane lipids such as cholesterol and gangliosides [10,11,12]. Tetraspanins associate with detergent-resistant membranes, as they were shown to float in sucrose gradients after cell lysis, indicating that tetraspanins assemble tetraspanin-enriched micro-domains (TEMs). Albeit TEMs and lipid rafts have been considered as seperate membrane micro-domains, it cannot be excluded that they cooperate [9]. Tspans have been shown to interact with a plethora of transmembrane proteins, including integrins, cell adhesion molecules and receptor tyrosine kinases. It is, therefore, not surprising that Tspans are involved in multiple biological processes ranging from important functions in the immune system and infectious diseases to pro- and anti-tumorigenic activities. Several Tspans, including Tspan2, Tspan8 (also known as CO-029 or TM4SF3), CD9 and CD81, all members of the same subfamily [13], from here on referred to as Tspan8 subfamily and distinct from the TspanC8 subfamily, were associated with tumorigenesis [14,15,16,17] and metastasis [16]. Depending on the cellular context, CD9 was associated with pro- and anti-tumorigenic activities [16]. In contrast, Tspan8 was demonstrated to correlate with metastasis of hepatocellular carcinoma [15], melanoma [18,19] and pancreatic adenocarcinoma [20,21]. Albeit Tspan8 has been linked to formation of tumor exosomes [21] and association of Tspan8 with integrins and cell adhesion molecules may facilitate cancer cell motility, the underlying molecular mechanism linking Tspan8 to cancer metastasis is still unclear [16].

Over the past few years, evidence has increased that Tspans are involved in the regulation of transmembrane metalloproteases. In particular, ADAM10 was demonstrated to interact with all members of the TspanC8 family, which is characterized by the presence of eight cysteine residues in the LEL. We and others demonstrated that TspanC8 members promote trafficking of ADAM10 from the endoplasmic reticulum to the plasma membrane [22,23,24,25]. A similar role was proposed for Tspan12 [26], while Tspan3 might be involved in stabilizing ADAM10 at the plasma membrane [27]. 

ADAM10 has furthermore been shown to constitutively associate with the tetraspanins CD9, CD81 and CD82. Engagement of these tetraspanins by specific antibodies redistributed ADAM10 into tetraspanin-enriched membrane patches correlating with the induction of ADAM10-mediated TNF α-shedding on B lymphoid cells [28]. 

Interactions of ADAM17 with Tspans are less well described. CD9 was demonstrated to interact with ADAM17 on leukocytes, keratinocytes and endothelial cells [29,30,31,32]. In leukocytes and endothelial cells, interaction of ADAM17 with CD9 impaired TNF α-shedding [7,28].

ADAM17 is found to be overexpressed in a large set of tumors, including breast, colon and liver cancer, and has been linked to autocrine activation of epidermal growth factor receptor (EGFR) on tumor cells [33]. Furthermore, ADAM17 is able to promote tumor growth via induction of inflammatory processes, such as TNF α signaling and interleukin 6 (IL-6) trans-signaling. TNF α is released from the tumor micro-environment, but also directly from tumor cells [34]. ADAM17 can generate a soluble form of the IL-6 receptor (sIL-6R) from cells that express the membrane-bound IL-6R. On target cells, the IL-6/sIL-6R complex can bind to the signal-transducing subunit gp130 and induce intracellular signaling in the absence of membrane-bound IL-6R [35]. This process is termed IL-6 trans-signaling and we and others demonstrated that this process is crucial for tumorigenesis of several cancer entities, including liver [36] and colon cancer [37].

Given the widespread overexpression of both ADAM17 and Tspan8 subfamily members in different tumor types, we assumed a functional interplay of these proteins in tumor cells. Here, we show that ADAM17 associates with Tspans CD81 and Tspan8 at the plasma membrane. However, all three investigated Tspans (CD9, CD81 and Tspan8) regulate ADAM17 substrate availability by different mechanisms. In particular, Tspan8 promotes the ADAM17-mediated release of TNF α from HEK293T and colon cancer cells. We demonstrate that through the recruitment of ADAM17 into tetraspanin-enriched micro-domains, and mutual binding to ADAM17 and TNF α, Tspan8 enhances TNF α-shedding. Therefore, Tspan8 might promote tumorigenesis via an increase in ADAM17-mediated pro-inflammatory signaling. Interference with Tspan8-ADAM17 interaction might represent a novel approach to selectively target pro-inflammatory and pro-tumorigenic ADAM17 substrate release without affecting other ADAM17 substrates.

## 2. Materials and Methods

### 2.1. Cell Culture and Transfection

HEK293T, HuH7, HeLa, MDA-MB231 and Lox cells were maintained in DMEM (Sigma-Aldrich, Steinheim, Germany) supplemented with 10% FCS (Pan Biotech, Aidenbach, Germany). HT29, HepG2 and IGROV1 cells were maintained in RPMI (Sigma-Aldrich, Steinheim, Germany) supplemented with 10% FCS. All cells were cultured at 37 °C, 5% CO_2_ and 95% relative humidity.

### 2.2. Generation of Stable Inducible Tetraspanin-Overexpressing HEK293T Cell Lines

In order to allow tetracyclin-inducible expression of Tspans, cDNAs coding for myc-tagged Tspan8, HA-tagged CD9 or HA-tagged CD81 were cloned via AgeI/MluI into pTRIPZ vector, replacing the TurboRFP cassette and shRNAmir insertion site. HEK293T were transfected with the respective expression constructs using linear polyethylenimine (PEI; Polysciences, Hirschberg, Germany) and selected with 2 µg/mL puromycin (InvivoGen, Toulouse, France). Tetraspanin expression was induced by the addition of 80 µg/mL doxycycline (Sigma-Aldrich, Steinheim, Germany). 

### 2.3. Generation of Tspan8-Deficient Cells

sgRNAs targeting exon 4 of the human TSPAN8 gene, the first completely protein-encoding exon in the gene, were designed with the Zhang lab guide design resources (http://tools.genome-engineering.org; accessed on 16 February 2016). Two different sgRNAs with lowest expected off-target effects were selected and cloned into pLentiCRISPRv2 as previously described [38]. HT29 cells were transfected using PEI with sgRNA constructs and subsequently selected with 2 µg/mL puromycin. Subsequently, single-cell clones were generated by limited dilution and tested for Tspan8 deficiency by immunoblotting.

### 2.4. Flow Cytometry

HT29 cells grown to confluency were detached using citric saline buffer, and washed 3× in PBS (Sigma-Aldrich, Steinheim, Germany). Cells were then incubated in blocking buffer (PBS + 1%BSA + 10%FCS) for 10 min on ice. After washing 1× in PBS, cells were incubated on ice for 30 min in antibody staining solution containing anti-Tspan antibodies and live/dead stain. Cells were subsequently washed 2× in PBS and analyzed on a BD FACSCanto II flow cytometer (BD Biosciences, Heidelberg, Germany). Data were analyzed using BD FlowJo v10 software (BD Biosciences, Ashland, OR, USA).

### 2.5. Exosome Isolation

Exosomes were isolated from cryopreserved human serum by differential centrifugation at 10,000× *g* in a table top centrifuge and at 120,000× *g* in a Beckman ultracentrifuge (Beckman Coulter, Krefeld, Germany). All centrifugation steps were performed at 4 °C. Exosome pellets were resuspended in PBS and intact vesicles were verified by light scattering and subsequently boiled in 2× Lämmli buffer.

### 2.6. Gaussia Princeps Luciferase-Based Protein Complementation Assay

Luciferase complementation assay was performed as described previously [39]. Briefly, HEK293T cells were seeded at a density of 1 × 10^5^ cells/mL in 500 μL/well of a 24-well plate. The following day, cells were transfected with plasmid DNA. Cells always received a combination of two plasmids, one coding for the C-terminal part of Gaussia princeps luciferase and one coding for the N-terminal part of Gaussia princeps luciferase, each fused to a protein of interest or as an empty control. On the second day after transfection, cells were washed, lysed in 100 μL passive lysis buffer (#E1941, Promega, Mannheim, Germany) for 15 min at RT and then transferred, in triplicates of 20 μL each, to a white 96-well bioluminescence plate. Coelenterazine was added and bioluminescence was detected using a Glomax plate injector system (Promega, Mannheim, Germany). Normalized luciferase ratio (NLR) was calculated as described in [39]. An NLR above 3.5 was considered to indicate a specific interaction. All experiments were at least *n* = 3.

### 2.7. AP-Shedding Assay

Shedding of alkaline phosphatase (AP)-tagged substrates was carried out as specified by Inoue et al. [40]. In short, stably transfected or parental HEK293T cells were treated with doxycycline and transiently transfected with plasmids encoding AP-tagged ADAM17 substrates (TGFα, TNFα, IL-6R and IL-1RII). Cells were then stimulated with 500 nM phorbol 12-myristate 13-acetate (PMA, InvivoGen, Toulouse, France) to induce ectodomain shedding. In a variation of the experimental setup, histamine-stimulated shedding was investigated. To this end, cells were additionally transiently transfected with the histamine H 1 receptor and were stimulated with 10 µM histamine (Sigma-Aldrich, Steinheim, Germany) instead of PMA as previously described [40]. AP activities in the cell fraction and in the supernatant were measured by conversion of colorless para-nitrophenolphosphate substrate to yellow para-nitrophenol (Sigma-Aldrich, Steinheim, Germany) directly after substrate addition and one hour after substrate addition. To determine the relative amount of shed AP-substrate in comparison to the unstimulated controls (AP-release), the following calculations were used:

ΔOD405 = OD405_1h_ − OD405_0h_

AP-substrate in conditioned media (CM) (%) = $\frac{\Delta \mathrm{OD}405\;\mathrm{CM}}{\Delta \mathrm{OD}405\;\mathrm{CM}+\Delta \mathrm{OD}405\;\mathrm{Cell}}$

AP-release (%) = AP-substrate in CM under stimulated conditions − AP-substrate in CM under vehicle-stimulated conditions

### 2.8. ADAM17 Cell Surface Activity Assay

Cell surface activity of ADAM17 in HEK293T cells was measured using quenched fluorescent substrate peptides PEPDAB014 (BioZyme Inc., Apex, NC, USA). 1×10^6^ HEK293T cells were seeded into 6 cm-dishes and transfected with expression plasmids coding for the indicated tetraspanin. Then, 36 h post-transfection, cells were detached from the plate and seeded into a poly-L-Lysine-coated 96-well-plate at a concentration of 3.5 × 10^4^ cells per well. After attachment, cells were washed with PBS and incubated with 3 µM GI254023X (Sigma-Aldrich, Steinheim, Germany), 3 µM GW280264X (Iris Biotech, Marktredwitz, Germany) or 5 µM Marimastat (Sigma-Aldrich, Steinheim, Germany) for 30 min at 37 °C, as indicated, before the addition of 5 µM substrate peptide. Substrate peptide cleavage was measured in a 96-well-fluorescence-plate-reader (infinite^®^ 200Pro, Tecan, Männedorf, Switzerland) with excitation at 485 nm and emission at 535 nm in intervals of 30 s over two hours. The linear increase in fluorescence intensity was considered to be proportional to protease activity.

### 2.9. Enzyme-Linked Immunosorbent Assay (ELISA)

ELISA kits for sTNFα (DY-210) and sTNFR (DY-225) were obtained from R&D Systems (Wiesbaden, Germany) and the assay were performed from cell supernatants according to the manufacturer’s instructions.

### 2.10. Immunofluorescence

Cells were grown on poly-L-lysine-coated cover slips and fixed in 4% paraformaldehyde (Sigma-Aldrich, Steinheim, Germany) in PBS. Cells were permeabilized in 0.2% saponin (Sigma-Alrdrich, Steinheim, Germany) in PBS and quenched in 0.12% glycin and 0.2% saponin in PBS. Cells were incubated with primary antibodies (Appendix A) overnight at 4 °C and with Alexa Fluor^®^ 488-, Alexa Fluor^®^ 594- or Alexa Fluor^®^ 647-labeled secondary antibodies (1:200, Life Technologies, Darmstadt, Germany) for 1 h at room temperature. Coverslips were mounted with MOWIOL/DABCO with 1 µg/mL DAPI (Life Technologies, Darmstadt, Germany). Cells were then analyzed on an Olympus FluoView 1000 confocal microscope (Olympus, Hamburg, Germany). Co-localization was quantified by Mander’s co-localization coefficients determined by the JACoP plugin of ImageJ [41].

### 2.11. Red Fluorescent Protein Dimerization Assay

Generation of destination vectors with RFP-A and RFP-B located 3′ of the Gateway recombination site has been previously described [42]. HEK293T cells were seeded at a density of 2.5 × 10^4^ cells/mL in μ-dishes (Ibidi, Gräfelfing, Germany). The following day, cells were transfected with plasmid DNA. Cells always received a combination of two plasmids, one coding for RFP-A and one coding for RFP-B, each fused to a protein of interest or as an empty control. On the second day after transfection, cells were imaged on an Olympus FluoView 1000 confocal microscope.

### 2.12. Fluorescence Energy-Transfer (FRET) Microscopy

HEK293T cells were seeded in poly-L-lysine coated 8-well µ-slides (Ibidi). On the following day, cells were transfected with expression constructs coding for Tetraspanin-CyPet fusions as FRET donor and ADAM17-YPet or TNFα-YPet fusions as FRET acceptors using linear PEI. Control wells contained single transfections. Live cell fluorescent signals were recorded on an Olympus FluoView 1000 confocal microscope. Donor spectral bleed through (DSBT) and acceptor spectral bleed through (ASBT) were determined and subtracted from the recorded FRET signal, resulting in the precision FRET (pFRET) signal. FRET efficiency was calculated as with Qd and Qa as the quantum yield of donor and acceptor respectively and Ψdd and Ψaa as the collection efficiencies of the respective channel.

### 2.13. Stimulated Emission Depletion (STED) Super-Resolution Microscopy

Parental HT29 and Tspan8-deficient HT29 cells were grown on cover slips, fixed and stained as described above. AF594- and StarRed-labeled secondary antibodies (Appendix A) were used. Cover slips were mounted on glass slides and cell were imaged on a Leica DMi8 inverted fluorescent microscope equipped with an Abberior STEDYCON (Abberior, Göttingen, Germany) instrument with a STED depletion wavelength at 775 nm. Images were processed using ImageJ 1.52.

### 2.14. Sucrose Gradient Centrifugation

HT29 cells were lysed in TNE (25 mM Tris, 150 mM NaCl, 5 mM EDTA, protease inhibitors) containing either 1% Triton X-100 or 1% Brij-98. Sucrose gradient preparation, centrifugation and analysis of the corresponding fractions were performed as described previously [42,43].

### 2.15. Co-Immunoprecipitation Experiments

Cells were stimulated for 30 min with either 200 nM phorbol 12-myristate 13-acetate (PMA) or 1 µg/mL lipopolysaccharide (LPS, InvivoGen, Toulouse, France) and subsequently lysed for 15 min on ice in lysis buffer containing either Brij97 or digitonin (both Sigma-Aldrich, Steinheim, Germany). A total of 1 mg protein lysate was precleared with sepharose beads for 1 h at 4 °C and then incubated with primary antibody at 4 °C and constant agitation overnight. Protein complexes were isolated with protein G-agarose and subjected to SDS-PAGE and immunoblotting.

### 2.16. SDS-PAGE and Immunoblotting

Samples were lysed in RIPA buffer supplied with 50 mM NaF, protease and phosphatase inhibitors. Proteins were separated by electrophoresis on 10% SDS gels and transferred to PVDF or nitrocellulose membranes. Membranes were incubated with primary antibodies (Appendix A) overnight at 4 °C and with horseradish peroxidase-conjugated secondary antibodies at room temperature for 1 h. An ECL substrate kit was used for detection according to manufacturer’s instructions (Thermo Scientific, St. Leon-Rot, Germany).

### 2.17. Molecular Modeling

Due to the unavailability of CD9 and Tspan8 structure coordinate files, a model was generated using the structure coordinates of CD81 [44] (pdb entry: 5TCX) as template. Modeling was performed using the MODELLER interface of the UCSF Chimera package [45]. 

### 2.18. Gene Expression Meta-Analysis

RNA sequencing data were retrieved from The Cancer Genome Atlas database [46] using the RTCGA R package [47]. Data were processed using R 3.6.1 and the ggpubr R package [48] using the built-in Pearson linear correlation function. All data processing was performed from within RStudio ver. 1.1.456 software (RStudio Team (2020). RStudio: Integrated Development for R. RStudio, PBC, Boston, MA, USA URL http://www.rstudio.com, accessed on 14 November 2019).

### 2.19. Statistical Analysis

Data are represented as mean ± s.e.m. Normal distribution of all experimental data was tested using a Shapiro–Wilk test. Comparisons between two groups were performed by applying the Student’s *t*-test and if not normally distributed using the Mann–Whitney U test. For normalized samples, a significant deviation from 1 was tested using the one-sample Student’s *t*-test test. Comparisons between multiple groups were performed by applying one-way ANOVA with Bonferroni’s post-hoc test or if samples were not normally distributed with Kruskal–Wallis using Dunn’s post-hoc test. A *p* value < 0.05 was considered statistically significant. All statistical analysis were performed using R or GraphPad Prism software version 5 and 7 (GraphPad Software, San Diego, California USA)).

## 3. Results

### 3.1. Tetraspanins CD9, CD81 and Tspan8 Are Co-Expressed with ADAM17 in Tumor Cells

The Tspan8 subfamily comprises the members Tspan2, Tspan8, CD9 and CD81 [13]. We focused our analysis on the members Tspan8, CD9 and CD81, which were previously associated with pro-tumorigenic activities [16]. In silico assessment of transcriptomic data within the cancer genome atlas (TCGA) database indicated high expression of *CD9* and *CD81* in all analyzed tumor sets, while expression of *TSPAN8* was elevated in particular in hepatocellular carcinoma and colonic adenocarcinoma (Figure 1A). Expression of ADAM10 and ADAM17 was elevated in all tumor sets analyzed (Figure 1A). We did not find strong correlations between ADAM17 expression and the expression of Tspan8 family members (Appendix A). 

We analyzed expression of Tspans and ADAMs in selected tumor cell lines by flow cytometry and immunoblotting, respectively. High expression of Tspan8 was restricted to the colon carcinoma cell line HT29, while low to medium expression was detectable in IGROV-1, LOX and Huh7 (Figure 1B), reflecting the cancer genome data (Figure 1A). Expression of CD9 and CD81 was detectable in almost all cell lines (Figure 1B). Both ADAM10 and ADAM17 were expressed in all cell lines analyzed (Figure 1C). In the breast cancer cell line MDA-MB231, only weak ADAM17 was detected. Expression of ADAM10 was most pronounced in the hepatoma cell line HepG2 and the colon carcinoma cell line HT29. The pro-form of ADAM17 was present in all cell lines, while the mature form of ADAM17 was most strongly detectable in HepG2 and HT29 cells (Figure 1C). 

Given the strong expression of ADAM17 and Tspan8 in HT29, we assumed a functional association of both proteins. Using immunofluorescent staining, we confirmed that the two proteins co-localized at the plasma membrane of HT29 cells (Figure 1D). Interestingly, ADAM17 protein levels were slightly, but non-significantly increased in exosomes isolated from peripheral blood of a small cohort of patients suffering from colorectal carcinoma (CRC). ADAM17 protein levels in exosomes were even higher in patients with advanced stage CRC (Figure 1E). Taken together, Tspan8 subfamily members and ADAM17 are co-expressed and potentially functionally linked in different tumor cells.

### 3.2. Tspan8 Physically Interacts with ADAM17 at the Plasma Membrane

We used different approaches to investigate a potential physical interaction between ADAM17 and Tspan8 family members. Using a split-luciferase complementation assay [39] we detected strong interaction of ADAM17 with Tspan8 and to a lesser extent with CD81, but not with CD9 when overexpressed in HEK293T cells (Figure 2A). We generated fusion constructs of ADAM17 and Tspans with the FRET-optimized fluorescent proteins YPet and CyPet [49], respectively, and expressed these constructs in HEK293T cells. We performed FRET microscopy on these cells and detected Tspan8/ADAM17 interaction in a perinuclear compartment, but most prominently at the plasma membrane (Figure 2B). Similar results were obtained in ADAM10/ADAM17-deficient HEK293T cells (Appendix A). In order to confirm these result, we used a previously described RFP hetero-dimerization assay in transiently transfected HEK293T cells [50]. Proteins of interest were fused to fluorescently dim monomeric RFP variants. Upon protein–protein interaction, a fluorescently bright RFP hetero-dimer was restored (Figure 2C). We detected strong homotypic Tspan8 interaction exclusively in punctate structures of the plasma membrane (Appendix A), consistent with the establishment of TEMs. These structures partially overlapped with FITC-labeled cholera toxin B (CtxB) that binds to membrane micro-domains enriched in cholesterol and sphingolipids [51]. We confirmed strong interaction of Tspan8 to ADAM17 at the plasma membrane using RFP hetero-dimerization assay, while intracellular interaction was less pronounced (Figure 2C and Appendix A). Efficiency of the assay to detect intracellular interactions was confirmed by the detection of the previously described Tspan15/ADAM10 interaction [24] at endomembranes (Appendix A).

In order to assess if Tspan8/ADAM17 interaction also occurs under endogenous conditions, we isolated Tspan8 by immunoprecipitation from HT29 cells. In order to distinguish between direct association and secondary association, we used lysis conditions with Digitonin to disrupt secondary interactions or with Brij97 to preserve the tetraspanin-web [42,52]. In particular, the mature form of ADAM17 co-immunoprecipitated with Tspan8 under both lysis conditions (Figure 2D,E), consistent with a predominant direct interaction at the plasma membrane. The interaction was independent from exogenous stimulation of ADAM17 activity e.g., with phorbol 12-myristate 13-acetate (PMA) or lipopolysaccharide (LPS) (Figure 2E), suggesting that Tspan8/ADAM17 interaction is uncoupled from ADAM17 activity. Interestingly, β-actin also co-immunoprecipitated with Tspan8 (Figure 2D,E). Using fluorescent live cell microscopy of HEK293T cells expressing CyPet-fused Tspan8, we observed that Tspan8 co-localized with cortical filamentous actin, stained with LifeAct-mCherry (Figure 2F). This is consistent with previous studies that reported interaction of the actin cytoskeleton with TEMs [52].

Taken together, these data indicate that ADAM17 is physically associated with Tspan8, predominantly at the plasma membrane. 

### 3.3. Tetraspanins CD9, CD81 and Tspan8 Alter Substrate Specificity of ADAM17

We hypothesized that interaction of ADAM17 with Tspans alters its molecular function and potentially regulates substrate processing. We, therefore, assessed ADAM17 catalytic activity and its ability to cleave substrate proteins. We generated HEK293T cells stably expressing Tspan8 family members in a tetracyclin-inducible manner (Appendix A). It is important to note that maturation of ADAM17 was not altered by co-expression of Tspan8 subfamily members (Appendix A). We then used a TGF α-based quenched fluorogenic substrate peptide to analyze catalytic activity of ADAM17 at the plasma membrane of Tspan-expressing HEK293T cells (Figure 3A). Overexpression of CD9, as well as CD81 but not Tspan8, increased plasma membrane enzymatic activity (Figure 3B). Since the substrate peptide can be cleaved by both ADAM10 and ADAM17, we used specific inhibitors to determine the protease responsible for increased substrate cleavage. This increase was similarly impaired in the presence of the ADAM10-specific inhibitor GI254023X and the dual-specific ADAM10/ADAM17 inhibitor GW280264X, suggesting that CD9 and CD81 enhanced ADAM10 catalytic activity, while the catalytic activity of ADAM17 was not increased by either of the Tspans analyzed. 

We next assessed ADAM17 shedding activity using selected ADAM17 substrates with an alkaline phosphatase (AP) tag at the extracellular domain. We chose substrates that were expressed in the tumor subsets previously analyzed for Tetraspanin and ADAM expression (Appendix A). HEK293T cells inducible for tetraspanin expression were transiently transfected with the AP-fused ADAM17 substrates. Cells were either stimulated with PMA or cells were additionally co-transfected with histamine 1-receptor (H1R) and stimulated with 10 µM histamine, to achieve a more physiological shedding stimulus via a G protein-coupled receptor (GPCR) [40]. Relative AP activity was subsequently determined in the supernatant and on the cell surface 60 min after stimulation (Figure 3C).

Shedding activity was subsequently determined as previously published [40]. Proteolytic release of the epidermal growth factor receptor ligand transforming growth factor (TGF) α was not significantly influenced by the overexpression of either one of the Tspans. Shedding of tumor necrosis factor (TNF) α, however, was altered by the overexpression of all three Tspans. While CD9 and CD81 decreased TNF α shedding, it was increased in the presence of Tspan8 after both PMA- and histamine-stimulation. Release of interleukin 6 receptor (IL-6R) and interleukin 1 receptor 2 (IL-1RII) was significantly elevated upon overexpression of CD9 or CD81, respectively, only after histamine-stimulation (Figure 3D,E). Overexpression of Tspan8 lowered histamine-induced shedding of IL-1RII (Figure 3E). The increase in TNF α-shedding in the presence of Tspan8 was lost in ADAM17-/- HEK293T cells, confirming that the observed effect is ADAM17-specific (Figure 3F). These data indicate that Tspans influence substrate specificity of ADAM17-mediated proteolytic ectodomain release. 

Given the pronounced effect of Tspan8 on TNF α-shedding, we sought to explore the role of Tspan8 on ADAM17 substrate selectivity under endogenous conditions. To this end, we generated Tspan8-deficient HT29 cells using CRISPR/Cas9 technology by targeting exon 4 of human TSPAN8 (Appendix A). We observed a significant Tspan8 down-regulation in targeted HT29 cell pools (Appendix A). Single cell clones displayed a complete knock-out and were used for further analysis (Figure 3G–I and Appendix A). We did not observe morphological differences between parental or Tspan8-deficient HT29 cells, nor did we detect changes in ADAM17 maturation or expression upon deletion of Tspan8 independent of the cell line (Appendix A). We observed only minor changes in surface levels of CD9 and CD81 in Tspan8-deficient cells (Figure 3H). In order to analyze the effect of Tspan8-deficiency on ADAM17-mediated TNF α shedding, we stimulated HT29 with IL-1β to induce TNF α secretion as previously described [53]. IL-1β-dependent increase in p65 and p38 phosphorylation 15 min after stimulation was not altered in the absence of Tspan8 (Appendix A). Furthermore, IL-1β downstream signaling was not influenced by inhibition of neither ADAM10 nor ADAM17 via GI254023X and GW280264X, respectively (Appendix A). Consistent with the results obtained with HEK293T cells, Tspan8-deficiency led to a significant decrease in TNF α shedding, while proteolytic release of TNF receptor 1 (TNFR1) remained unchanged (Figure 3H). Inhibition with the ADAM10/17 dual-specific inhibitor GW280263X but not with the ADAM10-specific inhibitor GI254023X-impaired release of TNF α and TNFR1, indicating that in HT29 cells, both proteins are proteolytically released by ADAM17 (Figure 3I).

Taken together, we identified Tspan8 subfamily members as mediators of ADAM17 substrate selectivity.

### 3.4. Tetraspanins CD9 and Tspan8 Associate with TNFα at Different Subcellular Compartments

Given the pronounced effect of Tspan8 subfamily members on proteolytic TNF α release, we hypothesized that interaction of Tspans with TNF α might regulate its availability for proteolytic release. We, therefore, assessed co-localization of TNF α with Tspans in transfected HEK293T cells. CD9 displayed a pronounced co-localization with TNF α exclusively in a perinuclear region. In these cells, neither CD9 nor TNFα were present at the plasma membrane, suggesting that CD9 retains TNF α in perinuclear compartments (Figure 4A). Both CD81 and Tspan8 were predominantly localized at the plasma membrane, but to some extent also present at endomembranes. While Tspan8 displayed co-localization with TNF α, we did not detect an overlap of CD81 and TNF α signals (Figure 4A). In order to assess physical interaction, we performed FRET microscopy using HEK293T cells transfected with Tspans fused to CyPet and TNFα-YPet. While CD9 was predominantly localized perinuclearly, CD81 and Tspan8 were mostly found at the plasma membrane. We detected FRET signals for Tspan8/TNF α and CD9/TNF α interaction, but not for CD81/TNF α. Both Tspan8 and CD9 associated with TNF α in a perinuclear region. Additionally, Tspan8 displayed interaction with TNF α at the plasma membrane (Figure 4B). 

Taken together, our data indicate that while CD9 regulates TNFα shedding through retention in a perinuclear compartment, Tspan8 might enhance TNF α processing through dual interaction with ADAM17 and TNF α at the plasma membrane. 

### 3.5. Tspan8 Promotes ADAM17-Mediated TNF α Processing through Recruitment of ADAM17 to Tetraspanin-Enriched Micro-Domains (TEMs)

Using stimulated emission depletion (STED) super-resolution microscopy, we observed co-localization of Tspan8 and ADAM17 but also Tspan8 and TNF α in membrane patches that might represent TEMs (Figure 5A, upper panel). In line with this assumption, localization of ADAM17 and TNF α in membrane patches was impaired in the absence of Tspan8 (Figure 5A, lower panel). 

We, therefore, hypothesized that Tspan8 regulates ADAM17-mediated processing of TNF α through differential localization to membrane micro-domains. In order to address this question, parental and Tspan8-deficient HT29 cells were either lysed in Triton X-100 (Tx-100) or in Brij97 and subjected to differential density gradient centrifugation. While Tx-100 destroys TEMs, but preserves lipid rafts, Brij97 preserves both membrane micro-domains [42,52]. Intact membrane micro-domains float in the sucrose gradient. Detergent-resistant micro-domains (DRMs) include GPI-anchored proteins, caveolae, flotillins and, depending on the cell type, TEMs [52]. We, therefore, used flotillin-2 as a marker for floating DRMs. The transmembrane cell adhesion molecule E-cadherin and the associating β-catenin, known interacting partners of Tspan8 [16], as well as small amounts of β-actin co-migrated with flotillin-2 positive, Tx-100 resistant DRMs. The distribution of these proteins in Tx-100 resistant DRMs were not altered in the absence of Tspan8 (Figure 5B). Interestingly, TNF α also co-migrated with Flotillin-2 positive TX-100 resistant DRMs in both Tspan8-proficient and -deficient cells. However, while in the presence of Tspan8, we detected both a 25 kDa membrane-bound pro-TNFα form and a 15 kDa proteolytically processed form; only the 25 kDa form was present in TX-100 resistant DRMs in the absence of Tspan8 (Figure 5B,C). This is consistent with the absence of TNF α-directed proteolytic activity in these micro-domains. Indeed, we did not detect ADAM17 in these fractions. However, when cells were lysed with the TEM-preserving detergent Brij97, we detected both Tspan8 and ADAM17 in flotillin-2 positive fractions (Figure 5B). In particular, the catalytic-competent mature form of ADAM17 was present in these fractions. In contrast, in the absence of Tspan8, ADAM17 was absent from flotillin-2 positive fractions. These data suggest that Tspan8 recruits ADAM17 to DRMs that contain TNF α. Accordingly, also in Brij97-lysed cells, TNF α was present in 25 kDa and 15 kDa form in the presence of Tspan8, but only as the 25 kDa form in the absence of Tspan8 (Figure 5C). Taken together, Tspan8 is responsible for the recruitment of active ADAM17 to TNF α-bearing membrane microdomains.

Recently, the full-length structure of CD81 has been resolved by X-ray crystallography [44]. The authors identified a cholesterol-binding module formed by the four transmembrane helices and proposed that cholesterol binding switches CD81 from an open to a closed conformation, thereby releasing bound client proteins [44]. In order to explain the differential effects of Tspan8 subfamily members on ADAM17 shedding activity, we performed homology modeling of CD9 and Tspan8 based on the available crystal structure of CD81 (Figure 6A). Most of the residues forming the cholesterol-binding pocket are also present in CD9 and Tspan8. However, E219 in CD81 aligns to a glycine residue in both CD9 and Tspan8 (Figure 6A,B). Interestingly, E219A substitution in CD81 impaired cholesterol binding [44]. It is, therefore, likely that in both CD9 and Tspan8, binding of client proteins is independent from the cholesterol content of the plasma membrane compartment. This is consistent with our findings that ADAM17-Tspan8 interaction was found in cholesterol-rich membrane patches. Furthermore, the size of the variable region within the large extracellular loop (LEL) differs. While CD9 has the shortest variable region, Tspan8 possesses an extra 11 aa insertion (Figure 6B) compared to CD81. These variations in LEL length might contribute to the diversity of client protein binding. Taken together, we showed here Tspan8-mediated regulation of TNF a cleavage by ADAM17 (Figure 6C). LEL sequence variations and differential regulation by membrane lipids represent potential mechanisms of how tetraspanins specifically regulate substrate availability for membrane proteases.

## 4. Discussion

Members of the ADAM protease family are involved in a plethora of biological and pathological processes [1,2]. The family member ADAM17 has been linked in particular to chronic inflammatory processes such as rheumatoid arthritis [54,55]. These pathological conditions are driven by the inflammatory cytokines TNFα and interleukin 6 (IL-6). Cytokine-neutralizing therapeutics, such as etanercept, infliximab and tocilizumab, are in clinical use for chronic inflammatory disease [56]. ADAM17 is the major protease generating the soluble form of TNF α and the soluble IL-6 receptor (sIL-6R), which are considered to be pro-inflammatory [57], and the development of small molecule inhibitors targeting ADAM17 for the treatment of chronic inflammatory disease has been under intensive investigation. However, progress of these inhibitors beyond phase II clinical trials has failed so far due to their strong side effects [54]. This is not surprising given the multitude and diversity of ADAM17 substrates [58]. Therefore, a more selective targeting of ADAM17 activity is warranted which implies a more detailed knowledge of the molecular mechanisms underlying ADAM17 substrate selectivity.

In the present study, we demonstrate that tetraspanins of the Tspan8 subfamily are regulators of ADAM17 substrate selectivity. Using different cell biological and biochemical approaches, we identified CD81 and Tspan8 as novel interactions partners of ADAM17. We were not able to confirm the interaction of CD9 with ADAM17 in HEK293T cells, which had been published previously to occur in leukocytes [29]. This could indicate that CD9-ADAM17 interaction might depend strongly on the cellular context. The interaction of Tspan8 and ADAM17 was most prominent and predominantly detectable at the plasma membrane in patches that were enriched in cholesterol and sphingolipids. Using differential detergent lysis, we demonstrated that Tspan8-ADAM17 interaction is direct and not mediated via the tetraspanin web.

Overexpression of Tspan8 subfamily members in HEK293T cells differentially regulated release of the ADAM17 substrates TNF α, IL-6R and IL-1RII. While TNF α release was significantly promoted by Tspan8, it was impaired by CD9 and CD81. In contrast, only CD9 was able to enhance IL-6R shedding, while only CD81 promoted shedding of IL-1RII. Interestingly, by using Tspan8-deficient HT29 colon carcinoma cells, we observed that while Tspan8 promoted cleavage of the ADAM17 substrate TNF α, it did not alter ADAM17-mediated shedding of TNFR1. These data demonstrate that Tspan8 subfamily members are able to regulate ADAM17 shedding activity in a highly selective manner. 

We and others have previously demonstrated that tetraspanins of the TspanC8 subfamily promote trafficking of the close homologue ADAM10, thereby enhancing its catalytic activity at the plasma membrane [22,23,24]. Using tetraspanin overexpression in HEK293T cells and CRISPR/Cas9- and siRNA-mediated depletion, we did not observe any effect of Tspan8 subfamily members on ADAM17 maturation or trafficking. Furthermore, using quenched fluorogenic substrate peptides we demonstrated that Tspan8 subfamily members also do not alter ADAM17 catalytic activity at the plasma membrane. Therefore, the increase in shedding activity of ADAM17 towards selected substrates must be regulated via substrate availability. We, indeed, find that in HEK293T cells CD9 is able to completely retain TNF α in a perinuclear compartment through direct interaction, thereby impairing TNF α release. We detected direct interaction of Tspan8 with TNF α in a perinuclear compartment, but also at the plasma membrane. Using super-resolution microscopy and differential density centrifugation, we detected TNF α in TEMs. However, TNF α localization to TEMs was not dependent on Tspan8. We rather observed that Tspan8 mediated recruitment of mature ADAM17 into TEMs. We, therefore, propose a model in which cleavage of TEM-resident TNF α is enhanced through the Tspan8-mediated recruitment of ADAM17 (Figure 6C). Consequently, cleavage of other ADAM17 substrates such as TNFR1 that is not regulated by Tspan8 most likely occurs outside TEMs (Figure 6C). Therefore, localization of transmembrane proteins into TEMs might represent a safeguard mechanism to tightly regulate its proteolytic release. In this model, Tspan8 might function as a “protease-shuttle” and switch towards proteolytic release. It is plausible that iRhom2 is involved in this process, as it forms a stable complex with ADAM17 at the ER and the plasma membrane [59,60,61] and has been shown to be also involved in regulating ADAM17 substrate selectivity [62,63]. The potential alteration in cholesterol-binding and cholesterol-dependent regulation of tetraspanin conformation and client protein-binding might explain the differential regulation of ADAM17 substrate shedding by Tspan8 subfamily members. This selectivity is most likely increased further through the variability of the large extracellular loop. Interestingly, proteolytic release of ADAM17 substrates has been shown to be sensitive to cholesterol depletion [64,65,66].

ADAM17 and TNFα are associated with chronic inflammatory conditions predisposing to cancer formation and metastasis [55]. This concept has been shown in particular for colon cancer [37,55]. ADAM17 and Tspan8 are expressed in a multitude of cancers, including colorectal and hepatocellular carcinoma. It is plausible that beside its function in cell adhesion, migration and exosome formation, Tspan8 enhances primary cancerogenesis and metastasis through an increase in ADAM17-mediated pro-inflammatory signaling. It is possible that Tspan8 regulates additional ADAM17 substrates that are involved in intra- and extravasation, which were not part of this study, leading to enhanced metastasis of Tspan8-expressing cancer cells. Therefore, specific targeting of tetraspanin-ADAM17 interaction might represent a novel, more selective approach to target detrimental effects of ADAM17 in both chronic inflammatory and oncogenic disease, avoiding the severe side-effects observed with ADAM17 inhibitors.

## Figures and Tables

**Figure 1 cells-11-02683-f001:**
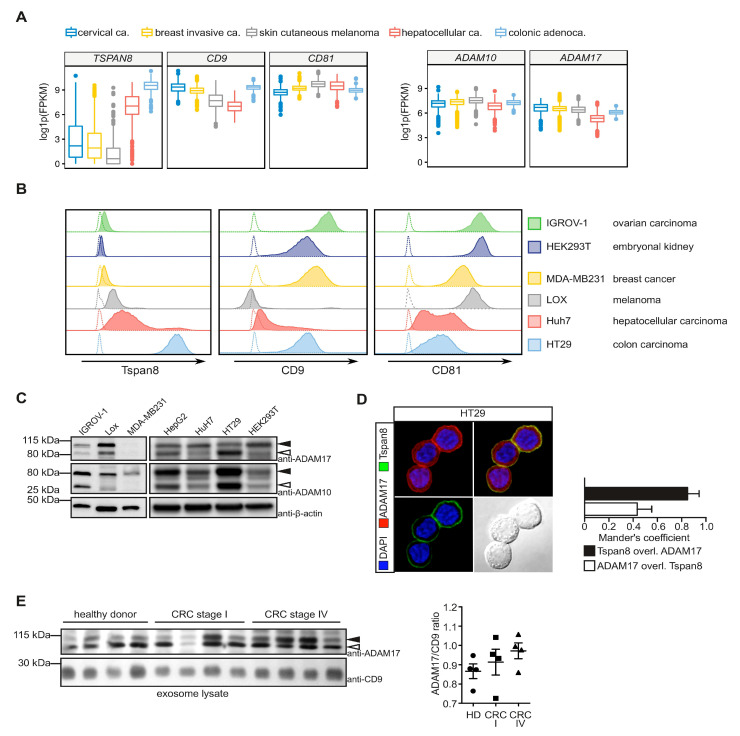
Expression of Tspan8 subfamily members and ADAMs in tumor cells. (**A**) mRNA expression of Tspan8, CD9 and CD81 is enhanced particularly in colonic adenocarcinoma, while expression of ADAM10 and ADAM17 can be detected in all tumor tissues analyzed. Expression data were retrieved from The Cancer Genome Atlas (TCGA). (**B**) Expression of Tspan8, CD9 and CD81 in the indicated tumor cell lines as assessed by flow cytometry. Dotted lines indicate unstained control. (**C**) Analysis of the indicated tumor cell lines for expression of ADAM10 and ADAM17 using SDS-PAGE and immunoblotting. (**D**) ADAM17 and Tspan8 co-localize at the cell surface of HT29 colon carcinoma cell line as assessed by immunofluorescence. (**E**) ADAM17 is enhanced in CD9+ exosomes isolated from the serum of patients suffering from colorectal cancer (CRC). Shown is the immunoblot of exosome lysates (left panel) and the densitometric quantification (right panel). HD: healthy donor, black filled arrow heads: ADAM10/ADAM17, pro-form, black open arrow heads: ADAM10/ADAM17 mature form. Data are represented as mean ± s.e.m., *n* = 3 independent experiments (**C**), One-way ANOVA on ranks with Dunn’s post-hoc test (E).

**Figure 2 cells-11-02683-f002:**
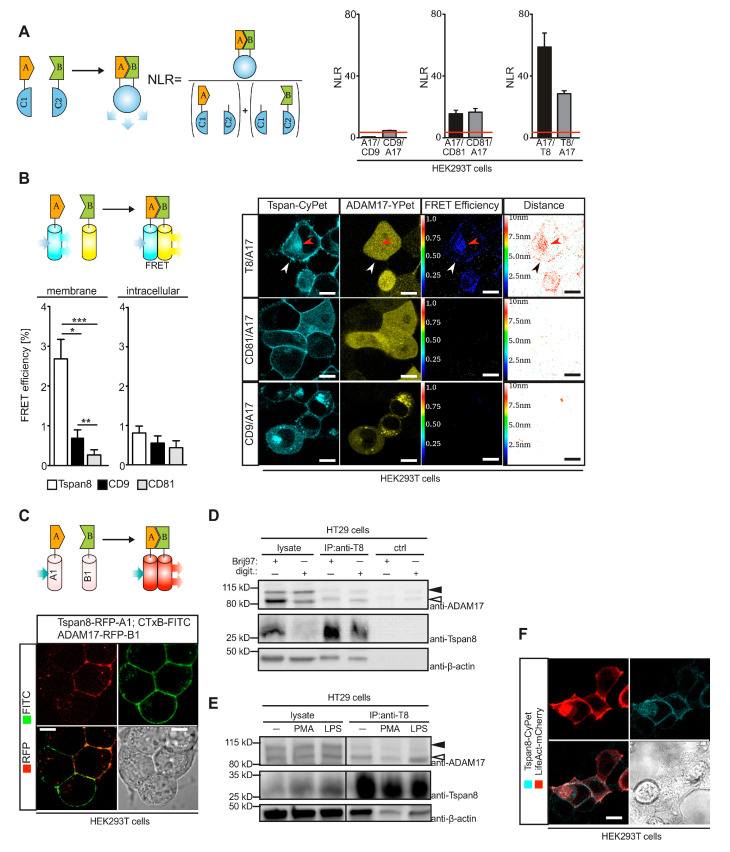
Tspan8 associates with ADAM17 predominantly at the plasma membrane. (**A**) Principle of split-luciferase assay. NLR denotes normalized luciferase ratio. ADAM17 specifically interacts with Tspan8 and CD81 but not with CD9 in overexpressing HEK293T cells, as assessed by split-luciferase assay. (**B**) Interaction of ADAM17 and Tspan8 can be detected at the plasma membrane and in perinuclear compartments using FRET microscopy of HEK293T cells expressing the indicated fusions of ADAM17 or Tspans with the indicated fluorescent proteins. (**C**) RFP heterodimerization assay confirms ADAM17-Tspan8 interaction at the plasma membrane in transfected HEK293T cells. (**D**) Endogenous ADAM17 co-immunoprecipitates with endogenous Tspan8 in HT29 colon carcinoma cells. digit.: digitonin (**E**) Interaction of ADAM17 with Tspan8 is independent of ADAM17 activation. HT29 were stimulated with 1 µg/mL LPS or 200 nM PMA and Tspan8 was isolated by immunoprecipitation and analyzed by SDS-PAGE for co-immunoprecipitating ADAM17. (**F**) Overexpression of CyPet-fused Tspan8 and mCherry-fused β-actin (LifeAct-mCherry) confirms co-localization of actin filaments and Tspan8 at the plasma membrane. Black filled arrow heads: ADAM17, pro-form, black open arrow heads: ADAM17 mature form. Data are represented as mean ± s.e.m. *n* = 3 independent experiments (**A**,**B**), * *p* < 0.05, ** *p* < 0.01, *** *p* < 0.001, Kruskal–Wallis with Dunn’s post-hoc test (**B**).

**Figure 3 cells-11-02683-f003:**
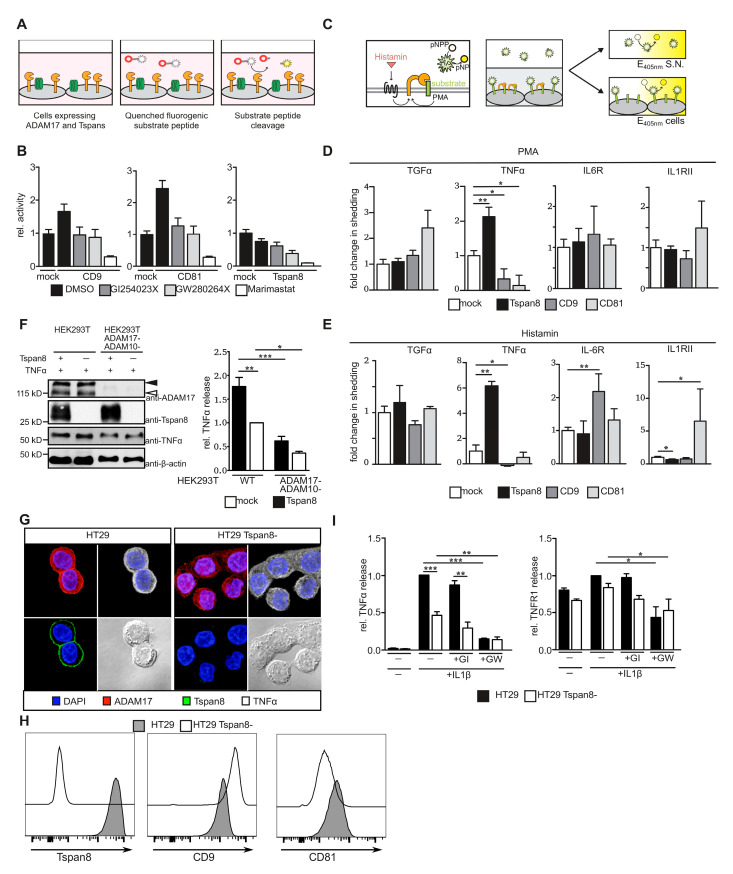
ADAM17 substrate shedding is selectively regulated by Tspan8 subfamily members. (**A**) Schematic of cell surface ADAM17 activity assay using a quenched fluorogenic peptide. (**B**) Cell surface enzymatic activity of ADAM17 is unaltered in the presence of CD9, CD81 or Tspan8, while cell surface enzymatic activity of ADAM10 is enhanced in the presence of CD9 or CD81. (**C**) Schematic of AP-shedding assay in HEK293T cells. Adapted from [40] (**D**) PMA-stimulated TNFα-shedding is enhanced in HEK293T cells overexpressing Tspan8 but impaired in HEK293T cells overexpressing CD9 or CD81. (**E**) ADAM17 substrate shedding is differentially regulated by Tspans in HEK293T cells overexpressing CD9, CD81 or Tspan8. Stable HEK293T cells were transfected with a plasmid encoding histamine 1 receptor and cells were stimulated with 10 µM histamine. (**F**) Tspan8-mediated increase in TNFα-shedding is dependent on ADAM17. Left panel: immunoblot of cell lysates using the indicated antibodies; right panel: TNFα release as assessed by ELISA. (**G**) Generation of Tspan8-deficient HT29 cells by CRISPR/Cas9 technology. Loss of Tspan8 was assessed by immunofluorescent staining. (**H**) Expression of CD9 and CD81 is barely altered in Tspan8-deficient HT29 cells as assessed by flow cytometry. (**I**) ADAM17-mediated TNFα but not TNFR1 shedding is reduced in Tspan8-deficient HT29 cells. Parental or Tspan8-deficient HT29 cells were stimulated with IL1β and proteolytic release of TNFα and TNFR1 was assessed by ELISA. Stimulated release of TNFα and constitutive release of TNFR1 is impaired by the dual-specific ADAM10/ADAM17 inhibitor GW280264X but not the ADAM10-specific inhibitor GI254023X. Black filled arrow heads: ADAM17 pro-form, black open arrow heads: ADAM17 mature form. Data are represented as mean ± s.e.m., *n* = 2–4 (**B**), *n* = 3–7 (**D**), *n* = 3–9 (**E**), *n* = 3 (**F**), *n* = 5–7 (**H**) independent experiments, * *p* < 0.05, ** *p* < 0.01, *** *p* < 0.001, Student’s *t*-test (**B**,**D**,**E**,**H**), one-sample Student’s *t*-test (**H**), one-way ANOVA with Bonferroni’s post-hoc test (**F**).

**Figure 4 cells-11-02683-f004:**
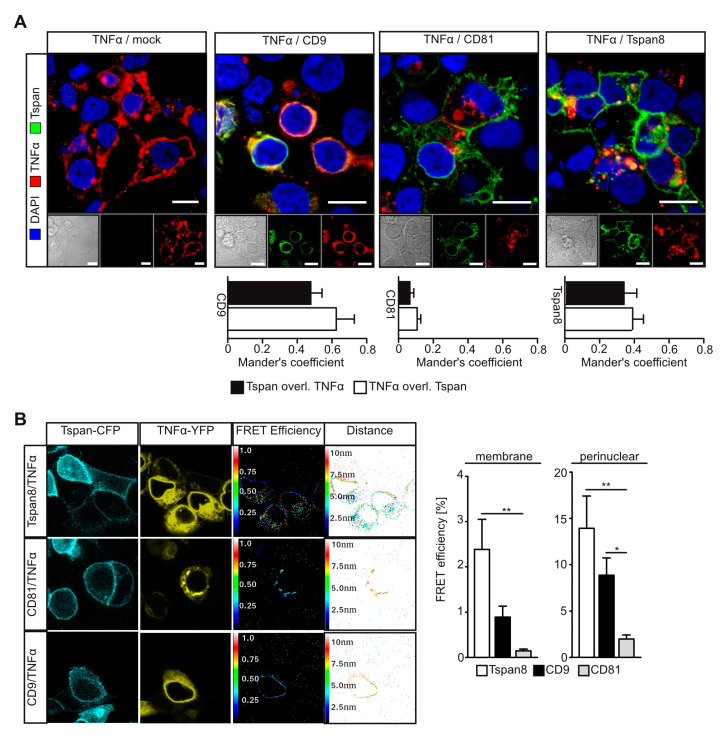
Tspan8 and CD9 interact with TNFα. (**A**) TNFα co-localizes with CD9 and Tspan8 at different subcellular compartments in transfected HEK293T cells as assessed by immunofluorescence. (**B**) Specific interaction of TNFα with the indicated Tspans in transfected HEK293T cells was assessed by FRET microscopy. Data are represented as mean ± s.e.m. *n* = 9 HPF images (**A**), *n* = 3 independent experiments (**A,B**), * *p*<0.05, ** *p* < 0.01, Kruskal–Wallis with Dunn’s post-hoc test (**B**).

**Figure 5 cells-11-02683-f005:**
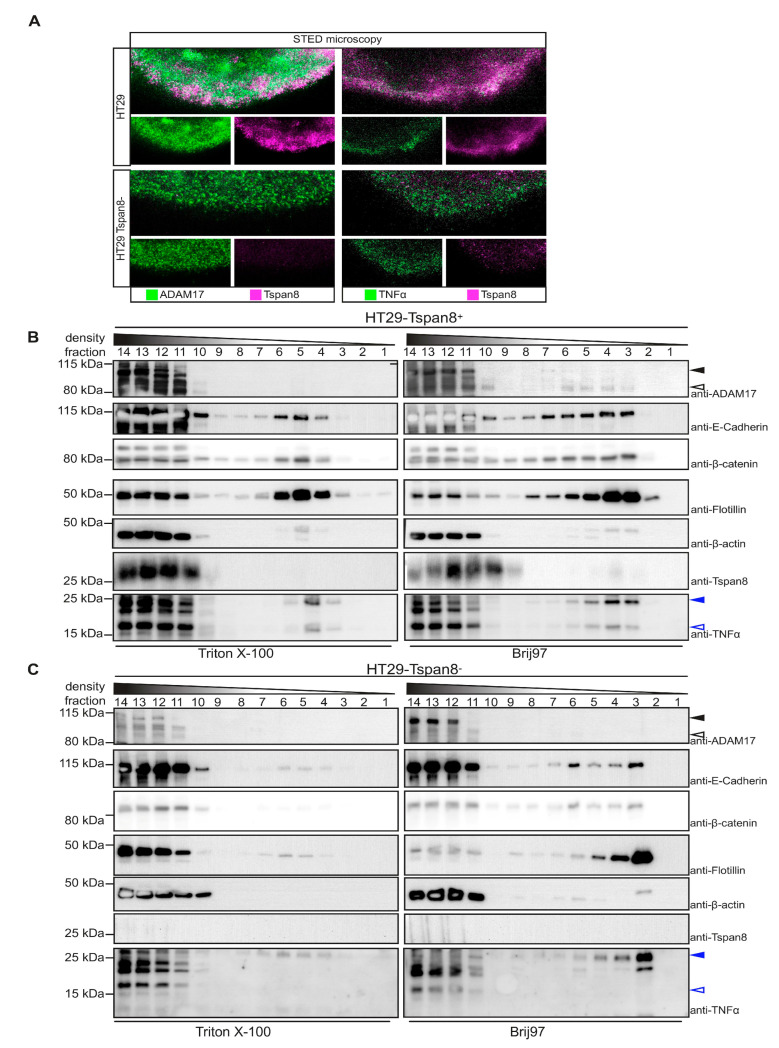
Tetraspanin 8 mediates TNFα shedding by ADAM17 in tetraspanin-enriched microdomains (TEM). (**A**) Stimulated Emission Depletion (STED) microscopy reveals presence of both ADAM17 and TNFα in Tspan8+ membrane microdomains. One representative out of three independent experiments is shown. (**B**,**C**) Tspan8 promotes recruitment of ADAM17 into tetraspanin-enriched microdomains. Parental (**B**) or Tspan8-deficient (**C**) HT29 cells were lysed with the indicated detergents and subjected to differential density centrifugation using a sucrose gradient. Flotillin served as a marker for lipid rafts in Triton X-100 lysed cells. Tetraspanin-enriched microdomains migrated at the same density as flotillin, as indicated by the presence of Tspan8 in HT29 cells. Black filled arrow heads: ADAM17 pro-form, black open arrow heads: ADAM17 mature form. Blue filled arrow heads: pro-TNFα, blue open arrow heads: processed TNFα. One representative out of three independent experiments is shown.

**Figure 6 cells-11-02683-f006:**
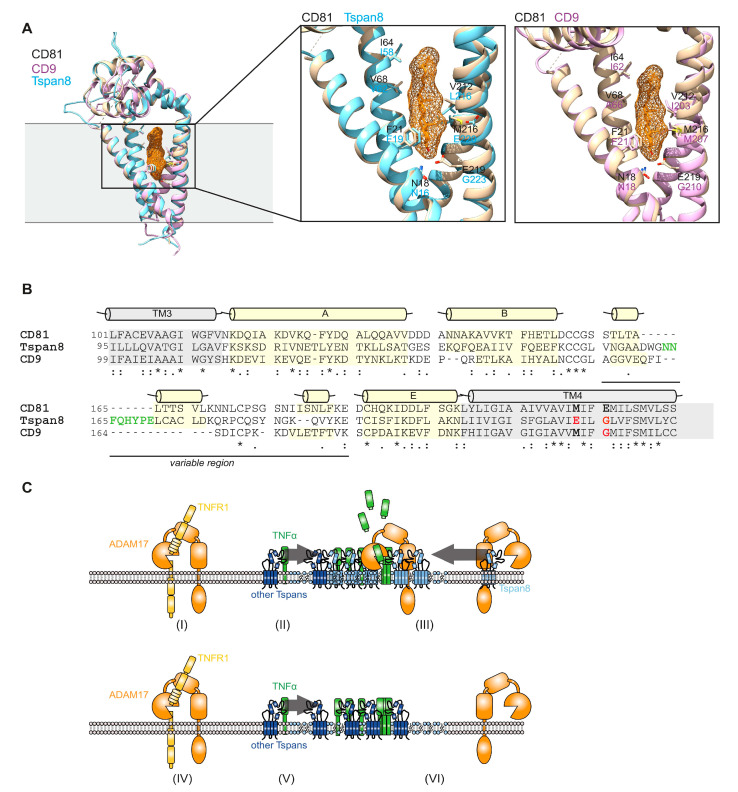
**Structural differences in the intramembrane potential cholesterol binding pocket might account for regulation of ADAM17 substrate availability.** (**A**) Homology model of CD9 and Tspan8 superimposed onto the crystal structure of CD81 highlight structural differences in the intramembrane potential cholesterol binding pocket. (**B**) Sequence alignment of the large extracellular loop (LEL) region between transmembrane helix 3 (TM3) and transmembrane helix 4 (TM4) reveals differences in sequence and length within the variable region. Amino acid insertions in the Tspan8 LEL are highlighted in green. Variations in amino acids of TM4 contributing to the cholesterol binding pocket in CD81 are marked in red. * denotes amino acid conservation, indicates strong similarity, indicates weak similarity. (**C**) Proposed model for Tspan8-mediated substrate selectivity of ADAM17. (I+IV) Substrates that do not bind to Tspan8 are cleaved outside tetraspanin enriched microdomains (TEMs). (II) Tetraspanins other than Tspan8 recruit TNF α into TEMs. (III) Tspan8 recruits ADAM17 into TEMs, enabling ADAM17-mediated release of soluble TNF α. (V) In the absence of Tspan8, TNF α is still present in TEMs, while (VI) ADAM17 cannot be recruited, therefore reducing TNF α cleavage.

## Data Availability

All data generated or analyzed during this study are included in this published article (and its Appendix A).

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
