# Peer review of "Tetraspanin 8 Subfamily Members Regulate Substrate-Specificity of a Disintegrin and Metalloprotease 17"

_cells, 2022, doi:10.3390/cells11172683_

Round 1

Reviewer 1 Report

The manuscript by Muller and co-workers describes and extensive and high-quality research on the interactions between ADAM17 protease and members of Tspan8 subfamily members, namely CD9, CD81 and Tspan8. In the course of this research, authors demonstrated that CD81 and Tspan8 regulate the catalytic activity and substrate specificity of ADAM17. These findings underline molecular mechanisms of ADAM17 regulation which might help to better understand its role in cancer. Moreover, targeting CD81-ADAM17 or Tspan8-ADAM17 interactions may appear as emerging strategy to combat inflammatory disorders, as it is known that ADAM17 is responsible for generating soluble form of pro-inflammatory IL6 cytokine. Overall this manuscript describes novel and very important findings in the field of ectodomain shedding and molecular signaling, and should be definitely published in Cells journal. The article is well written and easy to follow. The methodology is of high quality and the selection of approaches is ample. The conclusions are clear and fully supported by the data. In the Reviewer's opinion no further experiments are necessary; publish as it is. However, would it be possible to perform an in vitro experiments where ADAM17 and CD81 (or Tspan8) are co-incubated and the catalytic activity (and specificity) of ADAM17 is measured towards a set of IQF (or FRET) peptide substrates? This experiment can clearly demonstrate how CD81 (or Tspan8) affect ADAM17 activity and catalytic preferences. 

Reviewer 2 Report

The authors present a very nice study on protein co-factors that modulate substrate specificity of cell surface proteases. All conclusions are fully backed up by sufficient experimental data; using various complementing methods. The findings substantially add to our understanding o cell surface protease activity. In particular, the authors show co-expression, co-localization, and direct interaction of ADAM17 and TSP8 with TSP 8 modulating the sheddase activity profile of ADAM17. This is a very nice and complete study.